# A sensory integration account for time perception

**Alessandro Toso**[1], **Arash Fassihi**[1,2], **Luciano Paz**[1], **Francesca Pulecchi**[1], **Mathew E. Diamond**[1]*

**1** Cognitive Neuroscience PhD program, International School for Advanced Studies, Trieste, Italy,
**2** Department of Physics, University of California, San Diego, La Jolla, California, United States of America

* diamond@sissa.it

## Abstract

The connection between stimulus perception and time perception remains unknown. The present study combines human and rat psychophysics with sensory cortical neuronal firing to construct a computational model for the percept of elapsed time embedded within sense of touch. When subjects judged the duration of a vibration applied to the fingertip (human) or whiskers (rat), increasing stimulus intensity led to increasing perceived duration. Symmetrically, increasing vibration duration led to increasing perceived intensity. We modeled real spike trains recorded from vibrissal somatosensory cortex as input to dual leaky integrators–an intensity integrator with short time constant and a duration integrator with long time constant–generating neurometric functions that replicated the actual psychophysical functions of rats. Returning to human psychophysics, we then confirmed specific predictions of the dual leaky integrator model. This study offers a framework, based on sensory coding and subsequent accumulation of sensory drive, to account for how a feeling of the passage of time accompanies the tactile sensory experience.

## Author summary

The challenge to neuroscience posed by the sense of time lies, first and foremost, in the fact there do not exist dedicated receptors–the passage of time is a sensory experience constructed without sensors. In the present study, we have found that the perceived duration of a vibration applied to the skin increases not only in relation to actual elapsed time but also in relation to the intensity of the vibration. Our data uncover this robust relationship–"stronger is judged as longer"–in the psychophysical results both of human subjects and rats, indicating a general mechanism linking stimulus features to perceived time. We propose a computational model where the experience of the elapsed time accompanying a stimulus is generated when the activity of the sensory cortical neuronal populations encoding that stimulus is integrated by a downstream accumulator. We test the plausibility of the model by simulating the time percept that would emerge through integration of the neuronal firing of real spike trains recorded from the sensory cortex of rats receiving the vibratory stimulus. The close match of the model's prediction of perceived time to actual perceived time for the same stimuli supports the proposed sensory integration account for time perception.

**Data Availability Statement:** The authors confirm that all data underlying the findings are fully available without restriction. All relevant data are available through the following link: https://

figshare.com/projects/A_sensory_integration_
account_for_time_perception/63458.

**Funding:** Financial support was provided by the
European Research Council advanced grant
CONCEPT (https://erc.europa.eu; project 294498),
the Human Frontier Science Program (www.hfsp.
org; project RGP0015/2013), European Union FET
grant BIOTACT (project 215910), CORONET
(project 269459), and the Italian MIUR grant
HANDBOT (https://www.miur.gov.it/web/guest/
ricerca1; project GA 280778). The funders played
no role in the study design, data collection and
analysis, decision to publish, or preparation of the
manuscript.

**Competing interests:** The authors have declared
that no competing interests exist.

## Introduction

Every sensory experience is embedded in time, and is accompanied by the perception of the
passage of time. The coupling of the perception of the content of a sensory event and the time
occupied by that event raises a number of questions: Do these percepts interact with each
other? Do they emerge within separate neuronal populations? Which neuronal mechanisms
underlie the generation of two distinct percepts? By comparing and contrasting human and
rat psychophysics, the present study addresses these questions and thus aims to uncover gener-
alized mechanisms through which the brain represents the passage of time.

Two principal frameworks have been proposed to explain the neuronal bases of the feeling
of time in time scales of up to about 1 s: one framework posits a central clock, not connected
with any specific sensory modality [1] while a second framework posits that the cortical circuit
associated with each modality intrinsically encodes the passage of time for events within that
modality [2,3]. There are also mixed models that argue for the existence of a core timing struc-
ture that integrates cortical activity in a context-dependent way [4,5].

To determine to what extent the coding of time is embedded within the coding of the stim-
ulus itself, here we examine the relationship between the perceived features of a sensory event
and the perceived duration of that same sensory event. As a stimulus feature, we focus on the
intensity of tactile vibrations (quantified by their mean speed). The psychophysical experi-
ments reveal a systematic interaction between perceived intensity and perceived duration,
both in humans and in rats, leading us to propose that an early-stage sensory representation
provides common sensory input to two downstream integrators that generate two correspond-
ing percepts. To test this model, we use neuronal activity recorded from somatosensory cortex
of behaving rats to generate neurometric curves for perceived intensity and perceived dura-
tion, and derive a close match to observed psychophysical curves. From these findings, we pro-
pose a framework, general to humans and to rats, for the construction of the feeling of the
intensity of a stimulus and the time occupied by that same stimulus.

## Results

We carried out four experiments to investigate the effect of two tactile vibration features–
intensity and duration–on the two percepts directly connected to those features–perceived
intensity and perceived duration. Experiments 1 and 2 involve both human subjects, to whom
stimuli were delivered to the left index fingertip, and rats, to whom vibrations were delivered
to the whiskers on the right side of the snout (Fig 1A). Psychophysical experiments point to a
candidate mechanism for the generation of the percepts–two accumulators of sensory drive,
operating in parallel with percept-specific parameters of integration–and we probe the feasibil-
ity of the posited mechanism in Experiment 3 by generating intensity and duration neuro-
metric curves from recorded rat somatosensory cortex firing. Experiment 4 tests additional
predictions of the model in human subjects.

### Experiment 1: Interaction of vibration intensity and duration in a delayed comparison task

Each vibration was constructed by stringing together over time a sequence of velocity values,
$v_t$, sampled from a Gaussian distribution. We consider the stimuli as speed rather than velocity
since earlier work has shown that perceived intensity is mapped directly from vibration mean
speed [6,7]. The distribution then took the form of a folded half-Gaussian (right side of Fig 1B)
and the vibration can be considered a sequence of speed values, $sp_t$ (left side of Fig 1B). A single
vibration was thus defined by its intensity in units of mean speed in mm/s, denoted $I$

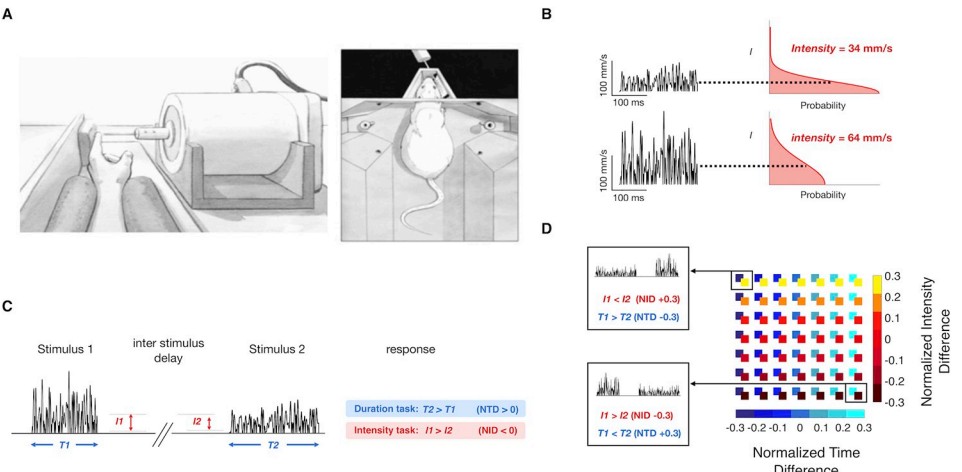

**Fig 1. Experiment conditions and stimulus parameters.** A) Left: Experiment setup, with the rat's whiskers in contact with the vibrating plate. Right: Experiment setup for the human, with the left index fingertip in contact with the vibrating probe. B) Schematic representation of the noisy vibrations delivered by the motor. The left side shows two traces of sampled speed over time, and the right side shows the folded half Gaussian distribution to which each sample corresponds. The distribution's expected value is shown for each trace. C) Delayed comparison trial structure. Each trial consisted of the presentation of two noisy stimuli, with specified durations and intensities, separated by an interstimulus delay. The response was deemed correct according to the task rule: to compare the relative durations (blue-shaded rule) or relative intensities of Stimulus 1 and 2 (red-shaded rule). D) Representation of all possible stimulus intensities and durations presented to the subjects in the delayed comparison task. Each square in the matrix is color coded according to the *NTD* and *NID* of the two vibrations presented. Selected *NID/NTD* combinations from the top left and bottom right of the matrix are illustrated.

(equivalent to the standard deviation of the Gaussian multiplied by $\sqrt{(2/\pi)}$). We consider *perceived intensity* to be the subjective experience related to objective intensity. Each stimulus was also defined by its duration in ms, *T*.

Experiment 1 employed a delayed comparison task, also known as the two interval forced choice task (Fig 1C). On each trial, subjects received two vibrations (Stimulus 1, Stimulus 2), separated by a fixed delay (500 ms for human subjects, 2 s for rats). The experiment was comprised of two distinct tasks: for *duration* delayed comparison, the subject had to judge which of the two stimuli was longer according to the relative *T* values (*T1 > T2* or *T2 > T1*). For *intensity* delayed comparison, the subject had to judge which of the two stimuli was of greater intensity (*I1 > I2* or *I2 > I1*). On trials when the parameter of interest was equal, the correct (rewarded) answer was assigned randomly. Each of 10 human subjects carried out both tasks, on different days, while individual rats were trained on a single task: 7 were *intensity rats* and 7 *duration rats*.

To constrain subjects to rely on working memory, we used a set of stimulus pairs referred to as the stimulus generalization matrix (SGM; Fig 2) in which any value of *I1* could be followed by a larger or smaller *I2* and any value of *T1* could be followed by a larger or smaller *T2* [6,8]. Since neither stimulus alone provided the information necessary for a correct choice, both stimuli had to be attended to and utilized to solve the task.

In each trial, the two stimuli could differ in *I*, in *T*, or both. To quantify stimulus differences, we designated two indices. The normalized *I* difference (*NID*), defined as (*I2 –I1*) / (*I2 + I1*), ranged from -0.3 to 0.3 for both humans and rats, while the normalized time difference (*NTD*), (*T2 –T1*) / (*T2 + T1*), ranged from -0.3 to 0.3 for humans and from -0.35 to 0.35 for rats. The stimulus set was constructed to present every possible combination of *NTD* and *NID* values during the session (Fig 1D). Subjects received the same stimuli whether the task was to

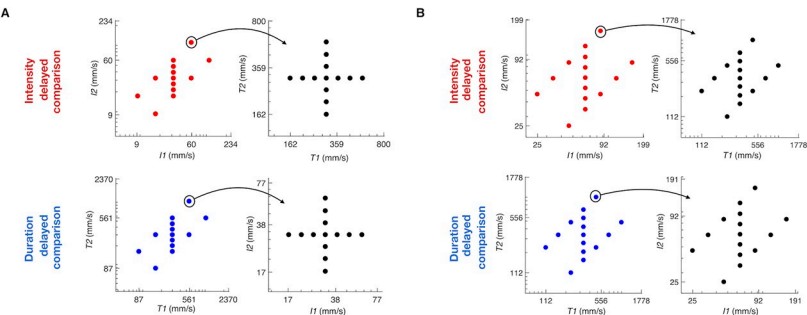

**Fig 2. Stimulus generation matrix.** A) Upper row: Matrices used for the intensity delayed comparison task. Lower row: matrices used for the duration delayed comparison task, for human subjects. Each trial's pair of task relevant feature values (*I* in the intensity task, *T* in the duration task) was drawn uniform randomly from the set of pairs scattered in the leftmost plots. Each trial's pair of task irrelevant feature values (*T* in the intensity task, *I* in the duration task) was drawn uniform randomly from the set of pairs scattered in the rightmost plots. B) Same as A, for rat subjects.

judge intensity or duration (see Fig 2 for the set of intensity and duration values). Thus, any resulting difference in performance of the tasks could not be attributed to differences in tactile input.

The upper plot of Fig 3A shows results from the duration delayed comparison sessions, with human data given as solid lines and rat data as dashed lines. The likelihood of judging *T2* > *T1* is plotted as a function of *NTD*, and the resulting steep curves report the capacity of human and rat subjects to extract the elapsed time. The similarity between the curves obtained from rats and human subjects demonstrates that the two species are approximately equally as proficient at discriminating stimulus durations.

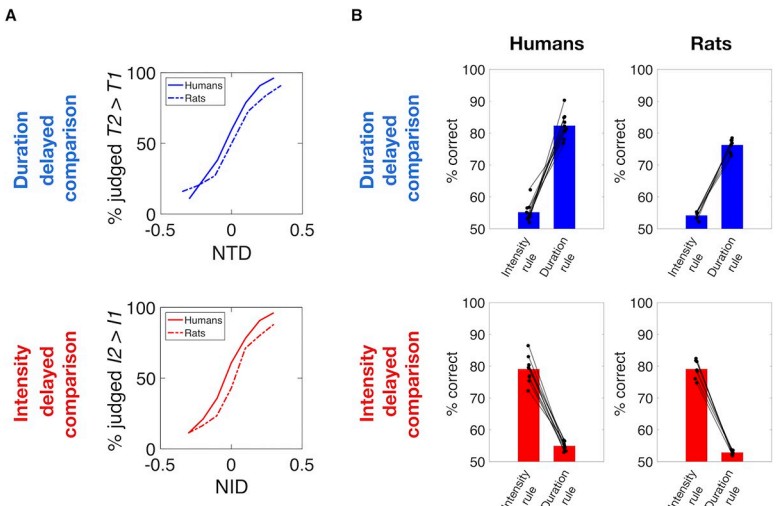

**Fig 3. Humans and rats extract two distinct percepts.** A) Psychometric curves averaged across all subjects as a function of *NTD* while averaging across all *NID* values (upper panel), and as a function of *NID* while averaging across all *NTD* values (lower panel). Solid lines show the choice probability for humans, while dashed lines show the choice probability for rats. B) Upper plots: Performance obtained by human subjects (left column) and rats (right column) in duration delayed comparison task. Bars on the left of each plot show the performance calculated according to the intensity rule (correctness according to stimulus intensity difference) revealing a consistent bias of the irrelevant feature on choice in both species. Bars on the right of each plot show the performance according to the duration rule, revealing similar performances in both species. Lower plots: Symmetrical analyses for intensity delayed comparison task, showing comparable performance and biases caused by the non-relevant feature on choice between the two species. In all plots, each line connecting a pair of dots represent single subjects.

By the same layout, the lower plot of Fig 3A show results from the intensity delayed comparison sessions. When the likelihood of judging *I2 > I1* is plotted as a function of *NID*, the resulting steep curves report the capacity of human (solid) and rat (dashed) subjects to extract stimulus intensity. Again, the two species show similar capacities in discriminating stimulus intensities.

Fig 3B shows the overall performance achieved by humans (left) and rats (right) in the two delayed comparison tasks. The left bar of each plot depicts the percentage of correct trials obtained when the subjects' performance is analyzed by the *intensity rule*, while the right bar depicts the correct percentage when the subjects' performance is analyzed by the *duration rule*. The upper panels show that the two species had similar performance (75–80% correct) in duration delayed comparison sessions when their choices were measured according to the duration rule. However, if their choices were measured according to the relative intensities of the stimuli, performance within the same stimulus set would remain above chance (about 55% correct). The small but significant bias according to the non-relevant stimulus feature means that in both species the higher-intensity stimulus, on average, tends to be judged as longer in duration. That is, stronger feels longer.

Similarly, the lower panels show that the two species had similar performance (75–80% correct) in intensity delayed comparison sessions when their choices were measured according to the *intensity rule*. However, if their choices were measured according to the relative durations of the stimuli, performance within the same stimulus would remain above chance (about 55% correct). Again, the small but significant bias according to the non-relevant stimulus feature means that in both species the greater-duration stimulus, on average, tends to be judged as higher in intensity. That is, longer feels stronger.

To assess the effect of vibration intensity on perceived duration in more detail, the upper panel of Fig 4A shows results from the duration delayed comparison sessions, with separate psychometric curves plotted for each value of *NID*. In both species, there was a pronounced

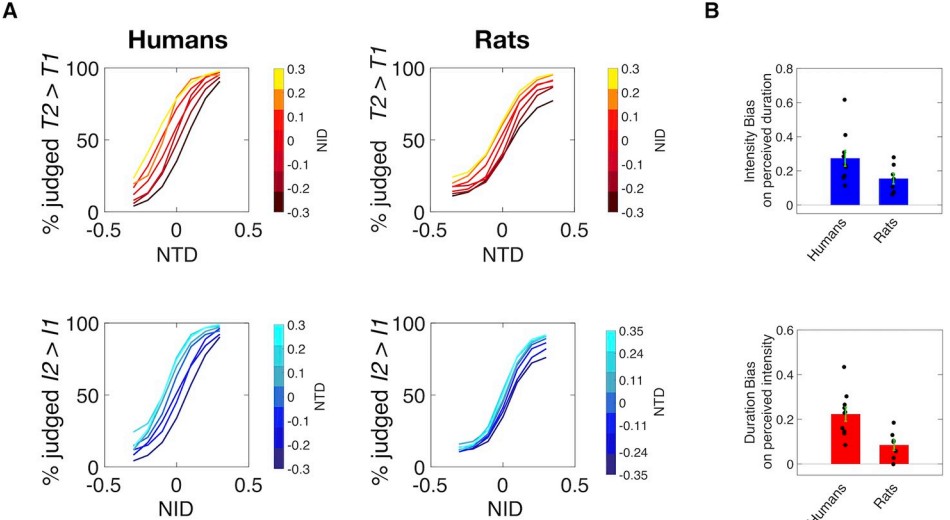

**Fig 4. Interacting stimulus features in delayed comparison.** A) Psychometric curves for 10 humans (left) and 7 rats (right) in the duration (top) and intensity (bottom) delayed comparison tasks. B) Upper plot: Bias caused by the non-relevant stimulus feature, intensity, in duration comparison. Lower plot: Bias caused by the non-relevant stimulus feature, duration, in intensity comparison. In all plots, dots represent single subjects, bars represent mean of biases across subjects, while error bars represent the standard error of the mean across all subjects. The median value of each bias was significantly different from zero (humans: $p = 0.002$ for both intensity and duration bias, rats: $p = 0.0156$ for intensity bias, $p = 0.032$ for duration bias, Wilcoxon signed rank test).

shift of the duration psychometric curves as *NID* grows from negative to positive (dark red to yellow), signifying that a greater value of *I2* relative to *I1* increased the likelihood of the subject judging *T2 > T1*. The lower panel reveals the effect of vibration duration on perceived intensity. The substantial shift of the psychometric curves as *NTD* grows from negative to positive (dark blue to cyan) signifies that, in both species, a greater value of *T2* relative to *T1* increased the likelihood of the subject judging *I2 > I1*. We quantified the bias in perception as the slope of the linear correlation relating the change in the non-relevant feature to the change in the *NTD* or *NID* value at which the subject judged Stimulus 1 and Stimulus 2 as equivalent (the point of subjective equality, PSE). In humans and rats, both in the duration task (Fig 4B, upper panel) and in the intensity task (Fig 4B, lower panel), the median value of bias was significantly above zero (humans: $p = 0.002$ for both intensity and duration bias, rats: $p = 0.0156$ for intensity bias, $p = 0.032$ for bias, Wilcoxon signed rank test, S1 Fig). Median values reveal a greater influence of the irrelevant feature in humans than in rats.

Two observations point to the shifts of psychometric curves as a perceptual rather than a decisional phenomenon. First, variations in the non-relevant feature affected choices in the same way whether applied to Stimulus 1 or Stimulus 2 (S2 Fig) even though Stimulus 1 is dissociated from any decisional process; the choice can be generated only after presentation of Stimulus 2. Second, the biases were better explained as a horizontal psychometric curve shift than a vertical shift (S3 Fig).

In short, the main finding of Figs 2 and 3 is that humans and rats readily extracted the perceptual feature required by the task, be it duration or intensity, but were biased by the non-relevant feature (intensity and duration, respectively).

## Experiment 2: Interaction of vibration intensity and duration in a direct estimation task

Delayed comparison (Experiment 1) involves a number of steps: Stimulus 1 encoding, storage in working memory, Stimulus 2 encoding, and comparison of current Stimulus 2 to the memory of Stimulus 1. Interaction between intensity and duration could occur during Stimulus 1 storage, recall, or during the Stimulus 2 versus 1 comparison. In Experiment 2, perception in human subjects was measured through *direct intensity estimation* and *direct duration estimation* (Fig 5A). If an intensity/duration confound were to persist in direct estimation, it would strengthen the argument that mixing emerges within the initial percept, before the percept is committed to or retrieved from working memory.

On a given trial, the subject received a single vibration, defined (as before) by *I* and *T*. A slider image appeared on the monitor 500 ms after the end of the vibration. By choosing the mouse-click position along the slider, the subject reported the perceived intensity of the vibration or else the perceived duration of the vibration. Scale normalization procedures are detailed in Methods. The slider did not display any landmarks, numbers, or ticks. To help subjects create two separate subjective scales, the orientation of the slider was specific to the task (e.g. horizontal for the intensity session and vertical for duration session), with task/orientation association randomized among subjects. The test stimulus set was comprised of 10 durations (linearly spaced from 80 to 800 ms in 80 ms steps) and 10 intensity values (linearly spaced from 9.6 mm/s to 67.2 mm/s in 6.4 mm/s steps). All 100 possible combinations of intensity and duration were presented in each session (Fig 5B).

Fig 5C, left panel, shows the results of direct duration estimation, averaged across subjects. Perceived duration increased not only with *T*, as expected, but also with *I* (designated by colors from dark red to yellow), confirming the main result from Experiment 1. From the same data set, Fig 5C, middle panel, plots the mean reported duration (pooling all presented durations)

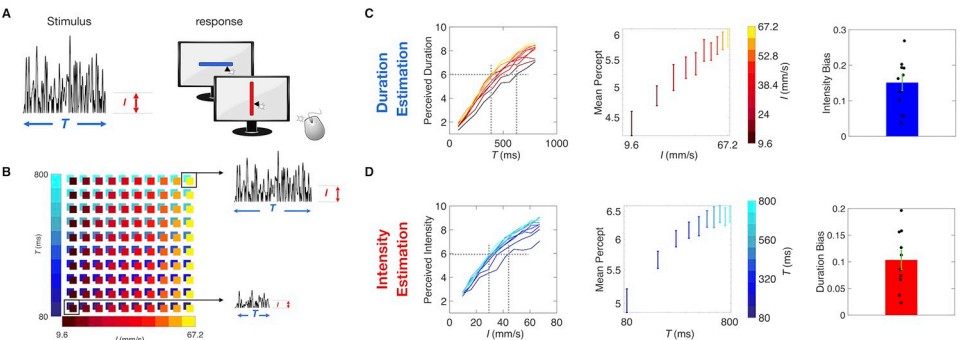

**Fig 5. Interacting stimulus features in direct estimation.** A) Experiment setup. 10 Human subjects received a single noisy vibration and reported perceived duration or intensity by mouse-clicking on a computer screen. B) Stimulus matrix. The vibration duration and intensity was randomly picked from the set of (*T*, *I*) combinations represented by the colored squares. Two sample stimuli from the upper right and lower left of the matrix are illustrated. C) Duration estimation results. The left plot shows the median perceived duration as a function of true duration. Middle plot shows how the mean percept, averaged across all values of *T*, changed with increasing values of *I* in log scale, for the duration estimation session. On the right, the intensity bias, calculated as the linear coefficient between mean perception and different values of *I* in log scale, across all 10 subjects. D) Intensity estimation results, following the same convention as panel C).

as a function of *I* (in log-scale), highlighting the intensity-induced bias. The rightmost panel of Fig 5C shows that the intensity bias, calculated as the linear coefficient of the fit between mean perceived duration and *I* in log scale, is significantly different from zero (Wilcoxon signed rank test, $p = 0.002$).

Fig 5D, left panel, shows the results of direct intensity estimation, averaged across subjects. Perceived intensity increased not only with *I*, as expected, but also with *T* (from dark blue to cyan). Thus, longer stimuli were perceived as stronger in direct estimation, as found earlier in delayed comparison. Again, the mean reported intensity increases with the non-relevant feature T (Fig 5D, middle panel); the duration bias (Fig 5D, right panel), calculated as the linear coefficient of the fit between mean perceived intensity and *T* in log scale, is significantly different from zero (Wilcoxon signed rank test, $p = 0.002$).

The interaction between the intensity and duration implies the existence of perceptual "invariants" in humans, and presumably also in rats. Considering the set of slider curves in the left plot of Fig 5C, a horizontal line at perceived duration 6 would intersect the yellow-to-dark red curves at a sequence of actual durations ranging from about 400 to 600 ms. Once adjusted for intensity, all such stimuli would be perceived as having the same duration. Notice that the spacing in time would decrease as intensity increases, highlighting that the iso-duration percept curves would not be linear: the effect of intensity on perceived duration saturates at the upper end of the intensity scale. Similarly, the slider curves in the left plot of Fig 5D imply intensity invariants. In rats, the existence of "iso-intensity percept curves" have been previously posited in rats and humans based on the influence of vibration duration on perceived intensity [7,9].

## Experiment 3: A sensory integration account for time and intensity perception

Next, we asked which computations the brain might use to construct the two percepts from a common stream of sensory input. Neurons in rat somatosensory cortex precisely encode instantaneous speed [7], and analogous coding mechanisms exist in primates [10]. Because the vibration was stochastic, no instantaneous value could provide an intensity estimate for the

entire vibration [11]. A subject could, in theory, achieve optimal performance in the intensity task by linearly integrating (summating) the output of *I*-coding neurons over the entire vibration and then normalizing the integrated value by elapsed time. The denominator of this normalizing operation–elapsed time–could itself be the basis of the estimate of stimulus duration. But this computation would not explain the observed perceptual confound between the relevant and the irrelevant stimulus features, inasmuch as the time counter in the normalization is conceived of as an independent term.

As an alternative, we posit that the brain constructs the percept of both stimulus duration and stimulus intensity by nonlinear accumulation of the sensory signal over time. Nonlinear relations between stimulus features *I*, *T* and percepts are hinted at by the psychophysical functions of Experiment 2, (left panels of Fig 5C and 5D; also see S4 Fig).

A renowned model of accumulation in perceptual decision-making is the leaky integrator, in which some form of input is summated across time, while the accumulator simultaneously diminishes by some proportion of its accumulated quantity [12]. Leaky integration of sensory input can be formulated as:

$$C\frac{d\gamma}{dt} = -\lambda\gamma + f(I_t, t) \tag{1}$$

where $\gamma$ is the percept, $f(I_t, t)$ is the external drive to the integrator, $\lambda$ is the leak rate and $C/\lambda = \tau$ specifies a time constant of integration. We now link the model to brain activity, hypothesizing that the leaky integrator's source of drive may be sensory cortex. To test the feasibility of this hypothesis, we ask whether neuronal firing from rat vibrissal somatosensory cortex (vS1) can be inserted into Eq (1) in place of the term $f(I_t, t)$, to generate neurometric functions consistent with the observed rat psychometric functions (see Methods, S5 and S6 Figs). If so, then the parameters of integration that optimize the similarity between neurometric and psychometric functions would be informative about the underlying brain mechanisms. Fig 6A shows raster plots of one neuronal cluster recorded from vS1 of an awake-behaving rat (see Methods) as it received vibrations. Stimulus duration was 1 s and plots are arranged according to vibration *I*, from 147 mm/s (top) to 42 mm/s (bottom). For present purposes, two forms of sensory drive emanating from vS1 are of interest: *I*-coding neurons and non *I*-coding neurons. The positive relation between *I* and firing rate is highlighted by the peristimulus time histogram at the bottom of Fig 6A; this was an "*I*-coding" neuronal cluster, as defined by the significant ($p < 0.05$) Spearman correlation coefficient between *I* and whole-stimulus average firing rate. Coding of *I* can also be negative, where firing rate decreases significantly as *I* increases.

The uppermost peristimulus time histogram (PSTH) of Fig 6B was generated from the averaged firing of all positive and negative *I*-coding neurons (n = 66). The lower PSTH of Fig 6B was generated from the averaged firing of non *I*-coding neurons (Spearman correlation coefficient $p > 0.05$ for each neuron; n = 57). Here, a small rise in firing rate at vibration onset is visible, but subsequent firing rate was not significantly correlated with *I*.

Could inputs like those shown in Fig 6B be accumulated to generate the neuronal substrates of the intensity and duration percepts? We implemented two leaky integrator models as an attempt to account for duration and intensity perception. The two integrators differ at three levels: the leak time constant $\tau$, the proportion of *I*-coding versus non *I*-coding neurons that are integrated, and the neuronal noise, which quantifies the variability in the firing pattern within the neuronal population that serves as the input to the integrator. To replicate the duration delayed comparison performance of one example rat, the input to the duration leaky integrator consists of 34% *I*-coding neurons (66% non *I*-coding neurons), accumulated with a time constant $\tau$ of 666 ms and a noise level of +/-3.1 standard deviations (see Methods for the details on the quantification of noise). Fig 6C shows how the accumulated quantity, $\gamma_{duration}$, of

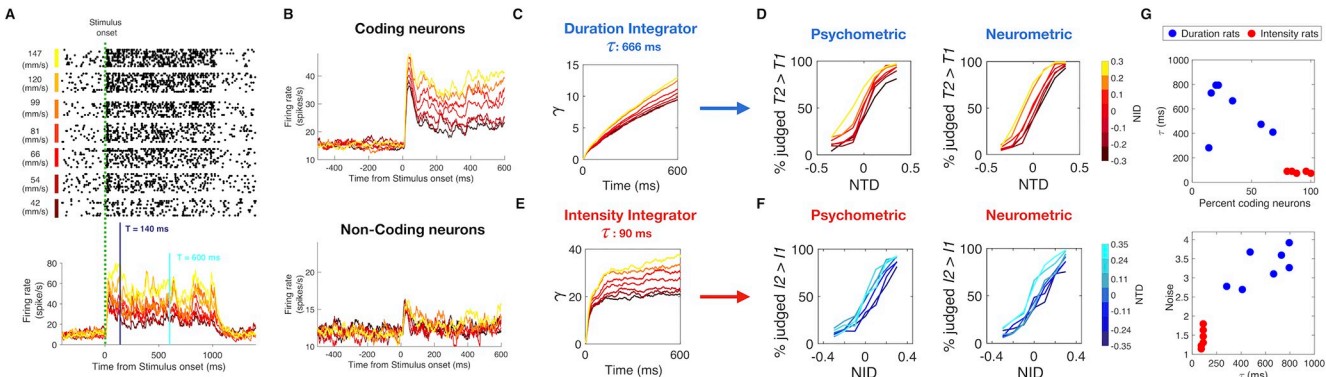

**Fig 6. Leaky integration of vS1 neuronal activity replicates psychophysical results.** A) Raster plots of one neuronal cluster recorded from vS1 of an awake-behaving rat as it received vibrations. Stimulus duration was 1 s and plots are arranged according to vibration *I*, from 147 mm/s (top) to 42 mm/s (bottom). Lower panel shows the peristimulus time histogram of the same neuronal cluster, sorted by vibration *I*. In order to replicate behavioral stimulus set, responses of individual neurons were measured from *t* = 0 to 7 different duration *T*, logarithmically spaced from a minimum of 140 ms to a maximum 600 ms. B) Upper plot: PSTH of all *I*-coding neurons (n = 66) sorted by *I*. Lower plot: PSTH of all non *I*-coding neurons (n = 57) sorted by *I*. Color scale for *I* as in A). C) Output γ of the duration leaky integrator as a function of time, obtained by integrating 34% of coding neurons and 66% of non-coding neurons with a time constant *τ* of 666 ms, and a noise parameter of 3.1 standard deviations. Color scale for *I* as in A). D) Comparison between the psychometric curves (left plot) and the neurometric curves (right plot) obtained for one example rat trained in duration delayed comparison. E) Output γ of the intensity leaky integrator as a function of time, obtained by integrating 90% of coding neurons and 10% of non-coding neurons with a time constant *τ* of 90 ms, and a noise parameter of 1.6. Color scale for *I* as in A). F) Comparison between the psychometric curves (left plot) and the neurometric curves (right plot) obtained for one example rat trained in the intensity delayed comparison task. G) Optimal values of the 3 parameters of the leaky integrator model obtained for each individual duration rat (blue dots) and intensity rat (red dots). The upper plot shows the percent of coding neurons versus *τ*. The lower plot shows *τ* versus the neuronal noise.

the duration leaky integrator grows over time for different vibration *I* values. The accumulated quantity builds up almost linearly, due to the long time constant, and increases modestly with stimulus *I*, due to the drive from *I*-coding neurons. The left and right panels of Fig 6D show the strong similarity between the psychometric curves and the simulated neurometric curves, respectively, obtained for this example rat (see S5 and S6 Figs and Methods for the model fitting procedure).

To replicate the intensity delayed comparison performance of one example rat, the input to the intensity leaky integrator consists of 90% *I*-coding neurons (10% non *I*-coding neurons), accumulated with a time constant *τ* of 90 ms and a noise level of 1.6 standard deviations. As seen in Fig 6E, the intensity leaky integrator, $\gamma_{intensity}$, saturates earlier than the duration leaky integrator, due to the short time constant. Moreover, the integrator grows more strongly with *I* than does the duration leaky integrator, a consequence of the predominant input from *I*-coding neurons. The left and right panels of Fig 6F show strong similarity between the psychometric curves and the simulated neurometric curves, respectively, obtained for this example rat. Psychometric/neurometric comparisons for all rats are given in S5 and S6 Figs.

Fig 6G shows the optimal values of the 3 variables of the leaky integrator model obtained for each individual duration rat (in blue) and intensity rat (in red). The optimal parameters for the two tasks were clearly separated in two clusters in all 3 dimensions. To replicate the behavioral results of the duration task, the sensory drive must be integrated with a long time constant (average *τ*, 592 ms; range 282–794 ms) compared to that of the intensity task (average *τ*, 83 ms; range 74–90 ms). If the duration percept, $\gamma_{duration}$, were to integrate the sensory drive with the time constant suitable for the intensity integrator, it would vary too greatly in relation to *I*, giving an unrealistically large intensity bias (S7 Fig, left plot). Symmetrically, if the intensity percept, $\gamma_{intensity}$, were to integrate sensory drive with the time constant suitable for the duration integrator, it would vary too greatly in relation to *T*, giving an unrealistically large duration bias (S7 Fig, right plot).

Moreover, the two integrators differ in the nature of their sensory drive. Duration delayed comparison neurometric curves replicate actual psychometric curves only when the sensory drive includes a low proportion of *I*-coding vS1 neurons with high neuronal noise, whereas intensity neurometric curves replicate actual psychometric curves only when the sensory drive includes a high proportion of *I*-coding vS1 neurons with low neuronal noise (Fig 6G). The duration leaky integrator's tolerance for non *I*-coding neurons and for noise implies that it is "open" to inputs unrelated to the vibration sensory code. This is consistent with the fact that time perception is a supramodal process; in the perceptual experience of an event, all sensory channels are connected with one unique feeling of time [13]. Furthermore, multimodal (audio-visual) stimuli are perceived as longer in duration than unimodal stimuli, suggesting the convergence of separate streams [14]. One possible interpretation of our data is that the duration leaky integrator normally accumulates neuronal activity from sensory areas besides vS1, reflected in the integrator's requirement to receive non *I*-coding activity with high noise within its sensory drive. On this basis, we predicted that the percept of duration (but not intensity) could be affected by input carried within a sensory modality other than that whose duration must be judged. Psychophysical experiments in human subjects support the prediction (see S8 Fig and S1 Text), revealing that an acoustic stimulus delivered together with tactile vibration is accumulated by the duration integrator but not the intensity integrator.

## Experiment 4: Dual leaky integrators

Having seen that the leaky integrator framework offers a plausible account for both percepts, when acting on sensory cortical spike trains, we next asked whether a single integrator is at work, one which operates with parameters tuned on each trial according to the current task (Model 1: Fig 7A). Alternatively, do two integrators, characterized by different parameter values, operate in parallel, with the subject retrieving the percept from the task-specified integrator (Model 2: Fig 7A)? To select between the two models, we designed Experiment 4, in which human subjects performed direct stimulus estimation, however the instruction cue indicating which stimulus feature to report was given to the subject either before or after stimulus delivery (Fig 7B). According to the single integrator Model 1 of Fig 7A, performance would be good on trials with cue delivery prior to vibration onset, inasmuch as the integration

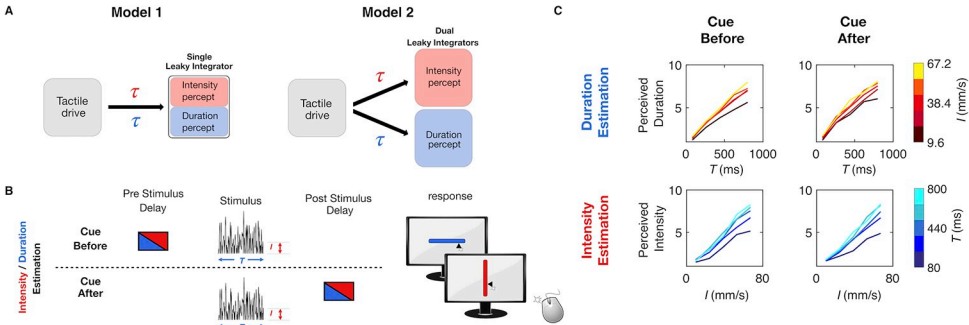

**Fig 7. Test of single versus dual integrators.** A) Alternative hypotheses for the leaky integration process underlying the construction of both intensity and duration perception. Model 1 is represented by a single integrator that receives tactile drive but switches between task-specific values for the parameter τ. Model 2 is represented by dual integrators that receive the same tactile drive. Each integrator has task-specific values for the parameter τ. B) Schematic representation of cue-before versus cue-after direct estimation experiment. On half the trials, the cue providing trial instruction (symbolized by red or blue box) was provided before the vibration (above dashed line), and on the remaining half, the cue was presented after the vibration (below dashed line). C) Comparison of median perceived duration (upper row) and median perceived intensity (lower row) when the cue was presented before (left column) versus after (right column) the vibration, for 8 human subjects. Time of cue did not affect estimation.

parameters could be correctly "pretuned" for the relevant feature. Performance would be lower with cue delivery after the conclusion of the stimulus inasmuch as the integration parameters could not be optimally pretuned prior to stimulus presentation. According to the dual integrator Model 2 of Fig 7A, performance would be nearly the same on trials with cue delivery after vibration offset versus cue delivery prior to vibration onset. This is because the two integrators operate in parallel, each with optimized parameters. The results, illustrated in Fig 7C, show that performance was not significantly affected by cue delivery time (Kruskal Wallis test: for duration estimation $p = 0.83$, Bayes Factor = 2.92, for intensity estimation $p = 0.75$, Bayes Factor = 2.88; see S9 Fig). Experiment 4 thus supports the dual integrator model.

## Discussion

### Rat tactile perceptual capacities

This study extends the range of high-level perceptual functions for which rats can serve as models. In Experiment 1, rats were able to perform a tactile *intensity* delayed comparison task with capacities similar to those of human subjects, replicating recent studies [6,7]. In Experiment 1, rats were also able to perform a tactile *duration* delayed comparison task, again with capacities strikingly similar to those of human subjects (Fig 3). Extracting stimulus duration and committing it to memory for future reference now enters for the first time the rodent perceptual repertoire.

### Intensity/duration confound reflects intertwined mechanisms

Having established rats as an appropriate target for investigation, the study turned to the problem of understanding how, from a single tactile stimulus, rats and humans can extract multiple percepts–the vibration's intensity together with the time it occupies. In both tasks, judgments were biased by the non-relevant stimulus feature–duration in the intensity delayed comparison task and $I$ in the duration delayed comparison task (Fig 4). As neither rats nor humans were able to generate independent representations of intensity and duration, a systematic perceptual interaction that transcends the species and the physical configuration of the receptors (fingertip and whiskers) must be at play. Experiment 2 replicated the perceptual interaction using a direct estimation task in human subjects, indicating that the interaction takes place regardless of whether the percept is manipulated in working memory (Fig 5).

Our results derive from the case in which the duration that must be measured is the elapsed time of the sensory stimulus itself. In conditions where the time to be judged lies between two discrete events—a start and stop signal, for example—the integration mechanisms remain to be worked out and the involvement of the dorsal striatum has been highlighted [15,16].

In earlier psychophysical work, an increase in perceived intensity with increased stimulus duration was reported in touch [7,17], audition [18–20] and vision [18]. The duration effect on perceived intensity was thought to arise from a temporal integration process: the sensory system summates input over time linearly, following Bloch's Law [21], or else nonlinearly [7,20,22]. Independently, an increase in perceived duration with non-temporal stimulus features has been reported in touch [23,24], audition [25] and vision [24,26]. However, the effect of stimulus features other than duration itself on perceived duration has been interpreted in a separate framework, which posits that an internal, central clock keeps track of time [1]. In this framework, the increase of perceived duration with increasing vibration $I$ would be an attentional phenomenon, where a stronger stimulus leads to an increase in arousal, resulting in augmented speed of the central clock [27,28].

By uncovering both perceptual confounds on the basis of a single set of stimuli (Figs 1D and 2), our experiments made it possible to configure perceived intensity and perceived duration within a unified framework. Opposing the view of two independent confounds, our experiments suggest that the influence of stimulus duration on perceived intensity and the influence of vibration $I$ on perceived duration constitute inextricable phenomena.

## Integration of sensory drive and alternative models

We propose leaky integration of one common source of sensory drive (Eq 1) as the mechanism underlying both sets of psychophysical data. In Experiment 3, we recorded vibration-evoked neuronal activity from vibrissal somatosensory cortex (vS1) of awake, behaving rats and inserted that firing as the sensory drive term of the leaky integrator formulation. After setting the parameters of sensory drive integration separately for the two tasks and then simulating choices by applying a decision rule to the leaky integrator's accumulated quantity, the resulting neurometric curves mimicked observed intensity and duration psychophysical curves (Fig 6).

The three parameter settings that optimized neurometric/psychometric overlap are informative about mechanisms. First, the leak time constant, $\tau$, was greater for the duration integrator (282–794 ms) than the intensity integrator (74–90 ms). These values fit with the intuition that the accumulated quantity underlying the perception of the passage of time must grow in a manner approaching linearity, while the accumulated quantity underlying the perception of vibration intensity must quickly reach a steady state. It is interesting that the intensity time constant is similar to that for the accumulation across touches of vibrissal kinematics-evoked activity in the formation of a texture percept [29,30], suggesting some common integrative mechanism or circuit underlying both percepts.

Second, the degree to which stimulus-unrelated neuronal activity is integrated differed for the two integrators. Intensity neurometric curves fully overlapped psychometric curves only when the leaky integrator input was comprised of 80–100% $I$-coding neurons, suggesting that the neuronal population underlying the final intensity percept gives more weight to synapses transmitting stimulus-specific information and tends to exclude non-responsive neurons. Whereas the tactile features of a multimodal event may be experienced distinctly from the concurrent visual or acoustic features, there is not a distinct sense of elapsed time for each sensory stream [14]. At the level of neuronal processing, this implies the convergence of diverse channels onto a single duration integrator, promiscuous to all inputs. Our study uncovered three correlates of this presumed convergence. First, duration neurometric curves matched psychometric curves only when the proportion of $I$-coding vS1 neurons in the sensory drive was reduced to 14–68%. Second, neurometric matched psychometric only when the noise level within the sensory drive was boosted from 1.2–1.7 (for the intensity neurometric) to 2.6–3.9 standard deviations. Third, the duration percept but not the intensity percept of tactile vibrations was influenced by the addition of acoustic noise alongside the trial (S8 Fig), again consistent with the expected multimodal convergence upon the duration integrator.

The hypothesis that perceived duration is achieved through leaky integration of stimulus-related and stimulus-unrelated sensory input is particularly relevant to the debate on models for the perception of time in the scale that ranges from tens of milliseconds up to a few seconds. Our results are in line with other work that assumes temporal integration processes behind the encoding of the passage of time [31,32] and are also in agreement with the idea that sensory-specific areas contribute to time perception [3,33]. They are harder to reconcile with the amodal central clock theory [1]. The "state-dependent networks" model [34] and the "striatal beat frequency" model [35] have not yet been challenged as to the dependence of time perception on non-temporal stimulus features. In the first type of model, time is encoded by the

evolving temporal pattern of activity of a recurrent neuronal network, so that almost any network could in principle represent the elapsed time without the need of an explicit representation of duration [34,36]. In the second type of model, time is encoded by a pattern of relative phases of different oscillators, thought to be present in both thalamic and cortical neuronal activity, and is read out by striatal neurons, which act as coincidence detectors. Both models would explain the influence of vibration $I$ on its perceived duration as an increase or decrease of the speed by which the activity of the connected neuronal population evolves in time or else by which the oscillators follow the pattern of their relative phases. Whether intensity-dependent modulation could be reliably implemented in these models is not known.

Beyond the integrative mechanisms acting on the stimulus within the single trial, another factor that can cause nonlinearity in the percept or choice is the recent history of stimuli [37,38].

## Integration timescales point to target regions and mechanisms

Temporal leaky integration of sensory information can be performed by a recurrent neuronal circuit [39–42]. Recurrent neuronal circuits have been widely used to explain decision-related neuronal activity in areas of the brain such as lateral intraparietal cortex (LIP) [43], premotor cortex [44], prefrontal cortex [45], and dorsal striatum [46,47]. The different level of leakiness for intensity versus duration could be achieved by a difference in the strength of recurrent connections in the network, and also on the different levels of background input [39,48,49].

Experiment 4, where human performance did not depend on preparatory knowledge about the percept to be reported, argues that the two percepts are generated through two computations that operate in parallel. We speculate that two different neuronal populations receive a common input, and the final state of these populations can be interrogated separately in order to produce the required judgment (Fig 6A, Model 2). The hierarchical ordering in the timescales of intrinsic fluctuations in the primate cortex, increasing from posterior to anterior [50,51], suggests that duration may be computed in a more anterior region with respect to intensity, perhaps at a downstream stage. Our rat experiments were limited to individuals reporting one percept, either duration or intensity. We speculate, however, that rats also simultaneously experience two distinct percepts, an assumption that would require discrete brain populations to simultaneously integrate sensory drive with different time constants. Primary somatosensory cortex is characterized by a short intrinsic timescale, and does not show temporal integration [7,11,29,50,52]. Intrinsic timescales for intensity integration may be found in primary and secondary vibrissal motor cortex (vM1 and vM2) and intrinsic timescales for duration integration may be found in farther anterior or medial regions. The convergence of tactile and acoustic input for the generation of the duration percept might occur in a population that processes sensory inputs from several modalities, such as premotor cortex [53] or dorsal striatum [54].

The two integrators are not a literal portrayal of a physiological mechanism, but are a characterization of some network property that participates in the conversion of the vibration sensory code to the conscious experiences of intensity and duration. The dual integrators constitute a unified framework inasmuch as both key features of the stimulus–the coding of vibration $I$ and elapsed time occupied by that stimulus–contribute to both integrators. While the framework put forward in this study cannot exclude the feasibility of all other models, it does create a set of predictions that can serve to alert us as to which network properties should be sought in future physiological work. For instance, the successful generation of neurometric curves to replicate both duration and intensity perception suggests vS1 as a common input, a hypothesis that could be directly tested by optogenetic control over vS1. Our model also makes

the straightforward prediction that the neuronal population implementing the readout of stimulus duration must be modulated by stimulus *I*.

## Methods

### Ethics statement

Human subjects were tested after giving their written informed consent. Protocols conformed to international norms and were approved by the Ethics Committee of SISSA (protocol number 6948-II/7).

14 male Wistar rats (Harlan, San Pietro al Natisone, Italy) were housed individually or with one cage mate and maintained on a 14/10 light/dark cycle. Daily access to water was restricted to promote motivation in the behavioral task, yet weight gain followed a standard Wistar-specific curve, indicating that the quantity of water obtained during training and testing was comparable to the ad lib quantity. After each session, rats were placed for several hours in a large, multistory enriched environment with other rats. Protocols conformed to international norms and were approved by the Ethics Committee of SISSA and the Italian Health Ministry (license numbers 569/2015-PR and 570/2015-PR).

### Human and rat subjects

Thirteen healthy human subjects (age range 22–35 yrs) were tested in the delayed comparison task. Only subjects that reached better than 70% performance in both intensity and duration delayed comparison (10 out of 13) were included in the analysis. The same 10 subjects were then recruited for the direct estimation tasks. All subjects were recruited on-line through the SISSA Sona System. (https://sissa-cns.sona-systems.com/). The number of subjects was 8 in Experiment 3 and 9 in Experiment 4. Individual rats were trained on a single delayed comparison task: 7 were *intensity rats* and 7 *duration rats*.

### Stimulus generation

Vibrations were generated by stringing together sequential velocity values ($v_t$) at 10,000 samples/s, taken from a normal distribution. Converting $v_t$ to its absolute value, $sp_t$, the distribution takes the form of a folded, half-Gaussian (see Fig 1B). A Butterworth filter with 150 Hz cutoff was then applied to yield low-pass filtered noise. The $sp_t$ time series for a given trial was taken randomly from among 50 unique sequences of pseudo-random values. Because stimuli were built by sampling a normal distribution, the statistical properties of an individual vibration did not perfectly replicate those of the distribution from which it was constructed. As a vibration's actual mean speed could not precisely match the distribution from which it was sampled, the assigned value may be considered "nominal" mean speed, referred to as *I*.

### Delayed comparison task for rats

Each trial began when the rat positioned its nose in the nose-poke (equipped with optic sensor) and placed its whiskers on the plate (Fig 1A). After a delay of 500 ms, Stimulus 1 was presented, characterized by intensity, *I1*, and duration, *T1* (Fig 1B). After the inter-stimulus delay of 2 s, Stimulus 2 (with *I2* and *T2*) was presented (Fig 1C). The rat remained in the nose-poke throughout both stimuli and could withdraw only when the "go" cue sounded at the end of the post-stimulus delay of 500 ms. Early withdrawal was considered an aborted trial and went unrewarded; it was not scored as correct or incorrect. After the go cue, the rat selected the left or right spout; reward location depended on the relative values of *I1* and *I2* for rats doing the intensity delayed comparison task, while it depended on the relative values of *T1* and *T2* for

rats doing the duration delayed comparison task. Incorrect choices went unrewarded. Trials with $I1 = I2$ or $T1 = T2$, according to the task, were rewarded randomly.

Rats learned the delayed comparison task by generalizing the comparison rule across the entire stimulus range, referred to as the stimulus generalization matrix (Fig 2), whereby neither stimulus alone provided the information necessary for a correct choice (1,2). Seven rats were trained to discriminate *T1 vs T2* and another 7 were trained to discriminate *I1 vs I2*. The stimulus range used during both duration and intensity delayed comparison task was from 112 to 1000 ms for stimulus duration, and from 25 to 160 mm/s for stimulus *I*.

## Delayed comparison task for human subjects

Experiments 1 and 4 employed a delayed comparison design. Stimulus 1 was characterized by intensity *I1* and duration *T1*. After the interstimulus delay of 1 s, Stimulus 2 (with *I2* and *T2*) was presented. Stimuli delivered to human subjects on the fingertip were the same as those used in rats except that the velocity values were halved. Each subject went through both an intensity and a duration delayed comparison session on different days. Subjects were verbally instructed to report which of the two stimuli was perceived as longer in duration or stronger in intensity, according to the behavioral task, by pressing the left (for Stimulus 1) or right (for Stimulus 2) arrow on the computer keyboard. They received visual feedback (correct/incorrect) on each trial through a monitor. A total of 1,456 trials were presented at each session.

The stimulus range used for intensity delayed comparison session was from 161 to 557 ms for stimulus duration, and from 9.28 to 110.36 mm/s for stimulus intensity. The stimulus range used for duration delayed comparison session was from 87 to 1034 ms for stimulus duration, and from 17.2 to 60 mm/s for stimulus intensity.

## Direct estimation task

The same human subjects that went through the duration and intensity delayed comparison task participated in the estimation task. Each subject went through both a duration estimation and an intensity estimation session, held on different days. Each session began with a training phase. In this phase, subjects received 40 stimuli, sampled randomly from the 100 possible stimuli (10 possible *I* values from 9.6 mm/s to 67.2 mm/s and 10 possible *T* values from 80 to 800 ms, linearly spaced), to become confident with the task and to sample the stimulus range. In the test phase, a single stimulus was presented, characterized by intensity, *I*, and duration, *T*. After a post-stimulus delay of 500 ms, a slider appeared on the screen. The slider did not present any landmark, ticks or numbers. The orientation of the slider was different between the two sessions, either vertical or horizontal. Subjects were instructed to report the perceived intensity or else the perceived duration of the vibration on a subjective scale, in which the extreme left/bottom position indicated a very weak or a very short stimulus, and the extreme right/top position indicated a very strong or a very long stimulus. The report was done by mouse-clicking on the chosen position along the slider. A total of 1,300 trials was presented at each session.

Five durations linearly spaced from 80 to 800 ms, and 5 *I* values from 9.6 to 67.2 mm/s were used. A visual cue, either a blue or a red square, was presented for 1 second, either before or after the delivery of the vibration. The orientation of the slider was kept horizontal for both intensity and duration estimation trials, so that the orientation could not be used as a cue for the trial type and subjects were forced to attend to the visual cue. A total of 1,300 trials was presented at each session.

## Analysis of human and rat delayed comparison data

To characterize the performance of the intensity delayed comparison task, we computed the proportion of trials in which subjects judged Stimulus 2 greater than Stimulus 1 on stimulus pairs characterized by a fixed *I1* (*I1* = 32 mm/s for human subjects, *I1* = 64 mm/s for rats) and different *I2* values, separately for each normalized time difference (*NTD*) value, defined as *(T2 −T1) / (T2 + T1)*. We fit the data with a four-parameter logistic function using the nonlinear least-squares fit in MATLAB (MathWorks, Natick, MA):

$$P(Stim2 > Stim1) = \delta + (1 - \delta - \lambda)\frac{1}{1 + e^{\left(-\frac{NID - \mu}{v}\right)}} \tag{2}$$

where *NID* is normalized intensity difference, (*I2-I1*) / (*I2+I1*), $\delta$ is the lower asymptote, $\lambda$ is the upper asymptote, $1/v$ is the maximum slope of the curve and $\mu$ is the *NID* at the curve's inflection point.

For the duration delayed comparison task we computed the proportion of trials in which subjects judged Stimulus 2 > Stimulus 1 on stimulus pairs characterized by a fixed *T1* (*T1* = 300 ms for human subjects, *T1* = 300 ms for rats) and different *T2* values, separately for each *NTD* value by fitting:

$$P(Stim2 > Stim1) = \delta + (1 - \delta - \lambda)\frac{1}{1 + e^{\left(-\frac{NTD - \mu}{v}\right)}} \tag{3}$$

where $\delta$ is the lower asymptote, $\lambda$ is the upper asymptote, $1/v$ is the maximum slope of the curve and $\mu$ is the *NTD* at the curve's inflection point.

In order to quantifying the bias of stimulus duration on the percept of stimulus intensity in the intensity delayed comparison task, we then computed a linear correlation between the *PSE* values fitted for different *NTD* values, and the actual *NTD* values. The additive inverse of the regression coefficient, was defined as *duration bias*. Symmetrically for the duration delayed comparison task, we computed a linear correlation between the *PSE* values fitted for different *NID* values, and the actual *NID* values. The additive inverse of the regression coefficient, was defined as *intensity bias*.

## Delayed comparison task: perceptual versus choice bias

Fig 4A reveals that both species show a pronounced shift in their psychometric curves on both duration and intensity discrimination tasks due to the task irrelevant feature, *NID* and *NTD* respectively. Horizontal and vertical shifts of the psychometric curve are frequently attributed to different steps in the cognitive process, a perceptual shift and a choice shift, respectively [55]. To quantify whether the shifts are better explained as purely horizontal or vertical we fit the following two models of decision probabilities:

$$P(Stim2 > Stim1) = (1 - p_L)s(\beta_p\Delta p + \beta_b\Delta b + \varepsilon_p) + p_L s(\varepsilon_L) \tag{4}$$

$$P(Stim2 > Stim1) = (1 - p_L)s(\beta_p\Delta p + \varepsilon_p) + p_L s(\beta_b\Delta b + \varepsilon_L) \tag{5}$$

where $P(Stim2 > Stim1)$ is the probability of choosing Stimulus 2 greater than Stimulus 1, $\Delta p$ and $\Delta b$ are the normalized differences of the relevant and irrelevant task features, respectively (i.e. *NTD* and *NID* for the duration delayed comparison task, and *NTD* and *NID* for the intensity delayed comparison task). $p_L$ is the probability of a lapse trial, $\beta_p$ is the linear weight that the task relevant feature has on choice, $\beta_b$ is the linear weight that the task irrelevant feature has on choice, and $s(x)$ is a logistic function, $\varepsilon_p$ is the constant perceptual bias, and $\varepsilon_L$ is the

**Table 1. Model comparison for the human and rat data.**

| Subject | Model Type | WAIC | pWAIC | dWAIC |
|---|---|---|---|---|
| Human | Perceptual bias | 15357±174 | 17 | 0 |
| | Choice bias | 15690±172 | 19 | 333±50 |
| Rat | Perceptual bias | 6146±143 | 42 | 0 |
| | Choice bias | 6181±133 | 39 | 35±101 |

The WAIC column shows the model's WAIC value with its corresponding standard deviation. The pWAIC column shows the model's effective number of parameters given the dataset. The dWAIC column shows the WAIC difference across each model and the one with the lowest WAIC value. The standard deviation of the dWAIC is smaller than that of each WAIC value, because each model's WAICs are correlated, as they are fit to the same dataset.

lapse trial constant choice bias. Both models in Eqs 4 and 5 assume that on each trial there is a probability $p_L$ that the choice will not be determined by the task-relevant features. A non-sensory error is called a lapse. On the remaining trials, choice is determined by a generalized linear model (GLM), specifically the logit link function, of the task relevant feature $\Delta p$. The two models differ in the role of the task irrelevant feature. In the model of Eq 9, $\Delta b$ is linearly combined with $\Delta p$ on the non lapsed trials, whilst in the model of Eq 10, $\Delta b$ goes through a separate GLM that determines the choice on lapsed trials. This means that the model of Eq 9 assumes that the task-irrelevant feature biases the effective percept yielding only a horizontal shift, whilst the model of Eq 5 assumes that the task-irrelevant feature biases choice on lapsed trials, yielding only a vertical shift.

We fit both models to the human and rat data using a Python software package PyMC3 [56]. In both cases, we pooled all the subjects together. We then compared the goodness of fit of both models using the Widely Applicable Information Criteria (WAIC, [57]). WAIC shows that the entire dataset is better explained by a perceptual bias (Eq 4) as compared to a choice bias (Eq 5). In the case of the humans, the normalized WAIC difference (ndWAIC which is equal to WAIC divided by its standard deviation) was equal to 6.62. In the case of the rats, ndWAIC = 0.34 (S3 Fig). A detailed comparison can be found in Table 1.

### Analysis of direct estimation task data

Results from the direct estimation task were analyzed taking into account two main factors. First is that the subjects tended to be more variable in their responses at the beginning of the session before setting a consistent subjective scale. In order to keep this variability from affecting our results, we calculated the mean response for each of the possible combinations of $I$ and $T$ and excluded outlier responses, those more than 1.5 SD displaced from the mean. Second was that not all subjects used the whole range of the slider; every participant set their minimum and maximum responses at a different position in the scale. In order to make each subject's subjective scale comparable, we used a min-max normalization algorithm:

$$Normalized\ x_n = 9\frac{x_n - min(x)}{max(x) - min(x)} + 1 \tag{6}$$

where $x_n$ is the non-normalized response on trial $n$, $x$ is the range of total responses, and *Normalized $x_n$* is the normalized response. We then multiplied by 9 and added 1, so that the normalized responses range from 1 to 10.

In order to estimate *duration bias* for intensity estimation trials in Experiment 3, we first computed the average normalized response across all possible $I$, for each $T$. We then computed a linear regression between the average normalized responses and stimulus $T$, and defined the regression slope as *duration bias*.

Similarly, in order to estimate *intensity bias* for intensity estimation trials in Experiment 3, we first computed the average normalized response across all possible *T*, for each *I*. We then computed a linear regression between the average normalized responses and stimulus *I*, and defined the regression slope as *intensity bias*.

## Electrode implantation and data acquisition

Trained rats (n = 2) were anesthetized with 2%–2.5% Isoflurane delivered with oxygen under controlled pressure through a plastic snout mask. They received an implant in the primary vibrissal sensory cortex (vS1), which was accessed by craniotomy, using standard stereotaxic technique (centered 2.8 mm posterior to bregma and 5.8 mm lateral to the midline). The multielectrode array (Tucker Davis Technologies) was inserted via micromanipulator. The extracellular activity of vS1 was sorted into 31 single units and 92 multiunits, as verified through the spike waveform and the refractory period observed in interspike interval histogram using a MATLAB-based software, UltraMegaSort 2000 [58,59]. In total, 123 neurons were recorded in 7 recording sessions. Part of the data set analyzed in the present study has been analyzed to different purposes in previous work [6].

## Neuronal population response

All analyses and statistical tests were done with custom codes written in MATLAB. We propose that the population activity of the vS1 that represents vibration *I* is integrated by downstream areas to produce the percepts of intensity and duration. An observer is imagined to make a perceptual choice based on the neuronal population integrated value, $\gamma$. The resulting choice, when plotted against stimulus difference (*NID* or *NTD*), is referred to as the neurometric curve. A high degree of similarity between the neurometric curve and the psychometric curve observed in the behavioral task supports the feasibility of the posited neuronal operation as a mechanism underlying behavior. In order to obtain an adequate number of trials per neuron per condition, we focused on trials with fixed and long duration and variable *I* values. In these sessions, stimuli lasted 200 ms, 600 ms, and in some sessions 1s. Only trials in which the stimulus duration was 600 ms or longer were used in the generation of neurometric curves.

In order to construct the neurometric curves from equivalent (*I*, *T*) values to those used in the psychometric analysis, pseudo random trials were created by measuring the responses of individual neurons from $t = 0$ to *T* (1 ms bin size) and averaged over equi-*I* trials. In order to have the same number of trials per combination of *I* and *T*, 200 resampled pseudo random trials per (*I*, *T*) stimulus class were generated by resampling [60]. The data set was then divided in half with the first half serving as the training set to estimate parameters of interest, and the second half serving to test how accurate the model with selected parameters predicts the observed psychometric curve.

## Leaky integration of neuronal firing

To quantify the coding properties of individual neurons, the Spearman correlation strength between stimulus *I* and the average response during the first 600 ms of the vibration was measured. Neurons with significant correlation ($p < 0.05$) were considered *I*-coding neurons (66 out of 123, 54%). The integrated neuronal response, $\gamma$, for each (*I*, *T*) stimulus class was calculated in 1 ms steps and $\gamma$ at time *T* (ms) was taken as input to the observer's choice. Neuronal activity drove the leaky integrator through a modified form of Eq (1):

$$C\frac{d\gamma}{dt} = -\lambda\gamma + f(r_t, n_t) \tag{7}$$

The external drive $f(r_t, n_t)$ to the integrator is:

$$f(r_t, n_t) = \frac{\sum_{i=1}^{N} rc_t^i + \sum_{j=1}^{M} rnc_t^j}{M + N} + n_t \tag{8}$$

where the first sum is the summation of the response of $N$ neurons randomly sampled from the $I$-coding neuronal population. The second sum is the response of non $I$-coding neurons, and $n_t$ is the neuronal fluctuation, sampled from a Gaussian distribution with 50 different values of variance and mean. The neuronal fluctuation within the sensory drive is generated from the Gaussian distribution with zero mean and standard deviation σ, $N(0, \sigma_t^2)$. The variance of the noise term at a given time-point is proportional to the average response of input channels pooled over all $I$-coding vs. non $I$-coding neurons at the same time-point ($R_t$).

$$R_t = \frac{\sum_{i=1}^{N} r_t^i + \sum_{j=1}^{M} rnc_t^j}{N + M} \tag{9}$$

$$\sigma_t = \sqrt{R_t n_t} \tag{10}$$

The proportion of coding vs. noncoding neurons was chosen among 50 different ($N$, $M$) combinations such that $\frac{N}{N+M}$ spanned 0 to 1 linearly. The leak time constant τ was chosen randomly among 50 values that spanned 50–800 ms linearly.

## Neurometric curves

For each combination of (τ, proportion coding neurons, noise level) parameters the neurometric curve is calculated as:

$$P(\text{choice}(stimlus2 > stimlus1)) = \delta + (1 - \delta - \lambda)P(Y_2 > Y_1) \tag{11}$$

where $\delta$ and $\lambda$ are the probability of upper and lower lapse (incorrect decision unrelated to perceptual processing) and are estimated from the psychometric curve of each subject. Integrator outputs for Stimulus 1 and Stimulus 2 ($\gamma_2$ and $\gamma_1$) are compared on a trial by trial basis (pseudorandom trial) for each ($I$, $T$) stimulus pair.

The neurometric curves should replicate two key features of the psychometric curves. The first is the bias, namely, the shift in the psychometric curve caused by the feature that should be exclude. This is quantified by the slope of the linear correlation relating the change in the non-relevant feature to the change in the $NTD$ or $NID$ value at which the subject judged Stimulus 1 and Stimulus 2 as equivalent–that is, the point of subjective equality (PSE). In the case of the neuronal analysis, we found that the noise in the bias measure (caused by neuronal response variability) could be diminished by computing the area under the neuromeric curve rather than the shift in PSE. The second key feature is the overall performance achieved by an observer who compares two stimuli on the basis of the integrated neuronal firing. We selected the values of leaky integrator parameters (τ, proportion coding neurons, noise level) such that the bias and performance resulting from the output of Eq (11) fell within 5 percent of the actual psychometric values that were meant to be replicated (S5 and S6A Figs). The neurometric curves constructed with the selected values of leaky integrator parameters produce similar performance and bias to those observed in behavior. For each selected value (S5 and S6A Figs, right, filled yellow dots), the neurometric curves were constructed using the test trials. The neurometric values for each ($I$, $T$) stimulus pair was then compared to the behavioral response. The parameter set that produces the least error was then used as the optimized integrator values (Figs 6D and 6G and S5 and S6B)

## Supporting information

**S1 Fig. Quantification of the perceptual biases.** Upper panel: Duration delayed comparison task. Each bar is the standard error, centered on the mean, of the psychometric curve PSE for each *NID* value across all 10 human subjects (solid) and all 7 rats (dashed), relative to the PSE for the *NID* = 0 condition. Lower panel: Same analysis for intensity delayed comparison task. The downward slanting distribution of data indicates that, for the duration task, PSE shifted to the left as *NID* grew while, for the intensity task, PSE shifted to the left as *NTD* grew.
(TIFF)

**S2 Fig. Effect of variations in the non-relevant feature whether applied to Stimulus 1 or Stimulus 2.** A) Upper row: Each bar is the standard error, centered on the mean, of the psychometric curve PSE for each *NID* value across all 10 human subjects, relative to the PSE for the *NID* = 0 condition, when varying the non-relevant feature of Stimulus 2 (left panel) or Stimulus 1 (middle panel), for the duration delayed comparison task. Right panel shows bias caused by the non-relevant stimulus feature, intensity, in duration comparison, for both conditions. Dots represent single subjects, while error bars represent the standard error of the mean across all subjects. Lower row: symmetrical analysis for intensity delayed comparison task. B) Same analysis as in B, for 7 rats in duration delayed comparison (upper plots), and 7 in intensity delayed comparison (lower plots).
(TIFF)

**S3 Fig. Characterization of the observed bias as horizontal versus vertical curve shifts.** Left column shows the analysis of the rat data; right column human data. Uppermost plot shows the in and out of sample deviance estimated by WAIC using the perceptual (horizontal) and choice (vertical) biasing models. Empty dots show the out-of-sample deviance (WAIC) of each model, the filled dots show the in-sample deviance (WAIC—2 pWAIC) of each model. The black bars show the WAIC standard deviation of each model. Gray triangle shows the model's WAIC difference, and the bar represents the WAIC difference's standard deviation. Standard deviation is smaller, due to correlations between the computations of WAIC for each model. The middle row shows the data in the duration delayed comparison task (dashed lines) along with each model's prediction (solid curves). The bottom row shows the data in the intensity delayed comparison task (dashed lines) along with each model's prediction (solid curves). The columns correspond to either the perceptual or choice bias models. Much better fit is obtained in all cases by the perceptual (horizontal) shift model.
(TIFF)

**S4 Fig. Non-linearities between percept and stimulus *T*, in direct estimation.** A) Duration estimation results. The two plots show the median perceived duration as a function of true duration in a linear-linear scale (left) and in a log-log scale (right), with each color denoting one *I*. In log scale, perceived duration increases linearly with stimulus *T*, suggesting a non-linear interaction between the two. B) Intensity estimation results. Left plot shows the median perceived intensity as a function of *I* in a linear-linear scale, with each color denoting one duration. Right plot shows the median perceived intensity as a function of stimulus *T* in a log-log scale, with each color denoting one *I*. In log scale, perceived intensity increases linearly with *T*, suggesting a non-linear interaction between the two.
(TIFF)

**S5 Fig. Parameter optimization for the duration leaky integrator model.** A) Leftmost plots show the leaky integrator parameters values that yielded a match of overall performance and intensity bias, for an example rat. Upper panel shows the values of $\tau$ and percent *I*-coding

neurons that yield leaky integrator performance replicating the actual performance of the rat (black dots). Lower panel shows the values of $\tau$ and percent *I*-coding neurons yield leaky integrator bias replicating the intensity bias of the same rat (black dots). Middle panel shows the parameters that gave a match in performance (blue dots) and intensity bias (orange dots) in the 3d parameter space. Yellow dots indicate the parameters values that produced a match in both features. Among those parameter values, the ones that minimized the difference between the choice of the rat and the choice of the ideal observer, for all *NTD* and *NID* values, were used to generate the neurometric curves (rightmost panel). B) Psychometric curves (upper row) and neurometric curves (middle row) for all individual rats.
(TIFF)

**S6 Fig. Parameter optimization for the intensity leaky integrator model.** A) Leftmost plots show the leaky integrator parameters values that yielded a match of overall performance and duration bias, for an example rat. Upper panel shows the values of $\tau$ and percent *I*-coding neurons that yield leaky integrator performance replicating the actual performance of the rat (black dots). Lower panel shows the values of $\tau$ and percent *I*-coding neurons yield leaky integrator bias replicating the duration bias of the same rat (black dots). Middle panel shows the parameters that gave a match in performance (blue dots) and duration bias (orange dots) in the 3d parameter space. Yellow dots indicate the parameters values that produced a match in both features. Among those parameter values, the ones that minimized the difference between the choice of the rat and the choice of the ideal observer, for all *NTD* and *NID* values, were used to generate the neurometric curves (rightmost panel). B) Psychometric curves (upper row) and neurometric curves (middle row) for all individual rats.
(TIFF)

**S7 Fig. Neurometric curves obtained with the opposite leaky integrator.** A) Neurometric curves obtained by integrating the sensory drive with the time constant suitable for the duration integrator, plotted as a function of NID values. B) Neurometric curves obtained by integrating the sensory drive with the time constant suitable for the intensity integrator, plotted as a function of NTD values.
(TIFF)

**S8 Fig. Test for integration of task-unrelated sensory drive.** A) Schematic representation of a trial with non-informative acoustic noise delivered through headphones. B) Bars denote mean performance on the duration and intensity tasks for trials with noise on and off, across 9 human subjects. Each dot represents a single subject's mean performance. Orange error bars are standard error of the mean across subjects. The presence or absence of noise did not affect accuracy (Kruskal-Wallis test, $p = 0.72$, Bayes Factor = 3.07 for the duration task, $p = 0.66$, Bayes Factor = 2.02 for the intensity task). C) Effect of acoustic noise on the bias caused by the task-irrelevant feature. For the duration task, the presence of noise reduced the bias normally caused by intensity (one sample, one-tailed Wilcoxon signed rank test, $p = 0.0273$). For the intensity task, noise did not affect the bias caused by duration (one sample, one-tailed Wilcoxon signed rank test, $p = 0.5$). Each dot represents a single subject's bias difference, whilst the bar represents the average across subjects. Error bars are standard error of the mean across subjects.
(TIFF)

**S9 Fig. Observed bias in Experiment 4.** Left plot: Bias caused by the irrelevant feature (*I*) in the duration estimation task. Each dot corresponds to a single subject, while the bar is the mean across subjects. Right plot: Bias caused by the irrelevant feature (*T*) in the intensity estimation task. Each dot corresponds to a single subject; the bar is the mean across all 8 subjects.

In both plots, orange error bars are standard error of the mean across subjects. Cue delivery did not affect the bias of the non-relevant feature (Kruskal Wallis test: for duration estimation $p = 0.83$, Bayes Factor = 2.92; for intensity estimation $p = 0.75$, Bayes Factor = 2.88). (TIFF)

**S1 Text. Discrimination task with supplementary auditory noise.**
(PDF)

## Acknowledgments

Marco Gigante, Fabrizio Manzino, and Erik Zorzin helped in instrumentation and realization of the experimental setup and stimuli. The Regional laboratory for advanced mechatronics, LAMA FVG (http://lamafvg.it) supported the design and construction of custom instrumentation. Fabrizio Manzino provided LabView programs for experiment control. Sara Sorella expertly helped train 4 duration rats. We are grateful to members of the Diamond lab for fruitful comments and discussions.

## Author Contributions

**Conceptualization:** Alessandro Toso, Arash Fassihi, Mathew E. Diamond.

**Data curation:** Alessandro Toso, Arash Fassihi, Francesca Pulecchi.

**Formal analysis:** Alessandro Toso, Arash Fassihi, Luciano Paz, Mathew E. Diamond.

**Funding acquisition:** Mathew E. Diamond.

**Methodology:** Alessandro Toso, Arash Fassihi, Francesca Pulecchi, Mathew E. Diamond.

**Writing – original draft:** Alessandro Toso, Arash Fassihi, Luciano Paz, Mathew E. Diamond.

**Writing – review & editing:** Alessandro Toso, Arash Fassihi, Luciano Paz, Mathew E. Diamond.

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
