## [Decision Letter · Decision Letter 0]

11 Nov 2020

Dear Professor Diamond,

Thank you very much for submitting your manuscript "A sensory integration account for time perception" for consideration at PLOS Computational Biology.

As with all papers reviewed by the journal, your manuscript was reviewed by members of the editorial board and by several independent reviewers. In light of the reviews (below this email), we would like to invite the resubmission of a significantly-revised version that takes into account the reviewers' comments. As you will see, the reviewers are generally positive about the manuscript. Unlike R2, I'm not concerned that the paper lacks an "unexpectedness factor"; strong theory-based experimental work is in my view a sufficient criterion for publication. In any event, I'm hopeful that you can adequately defend the theoretical contribution of this work.

We cannot make any decision about publication until we have seen the revised manuscript and your response to the reviewers' comments. Your revised manuscript is also likely to be sent to reviewers for further evaluation.

Sincerely,

Samuel J. Gershman

Deputy Editor

PLOS Computational Biology

Reviewer's Responses to Questions

**Comments to the Authors:**

Reviewer #1: The paper by Toso et al describes a novel computational insight into a very interesting, and puzzling, behavioral phenomenon – an intensity-duration “entanglement” (I will call it here IDE). The work integrates careful analysis of rodent and human behavior with single- and multi-unit activity from the cortex of rats performing the tasks. Having both behavioral and neuronal dynamics the authors were able to suggest a plausible model for the IDE, using dual leaky integrators with corresponding time constants for the intensity and duration integrators. they then confirmed specific predictions of the model with humans, which further strengthens its validity.

Beyond providing a plausible neuronal account for the IDE phenomenon, this work establishes an excellent model for investigating neuronal mechanisms of psychophysical behaviors in general. This of course depends on the ability to achieve in rodents behaviors that are comparable to those exhibited by humans, but, as this work shows, this is feasible and it is certainly worth the effort.

This is an excellent study and I don’t have any concern about its validity or value. I have, though, two suggestions that the authors may consider to use.

1. I was initially confused by the terminology. The authors use the term intensity to describe the mean speed. Intensity is not a quantitative term – it does not have units. Speed on the other hand is, and the authors in fact use speed units to describe the intensity variable. I suggest to seriously consider “speed” instead on “intensity” throughout the paper. I am aware to the possibility that I have missed an important consideration here, and I thus leave it to the decision of the authors.

2. a somewhat related issue. The results may indicate that there is some perceptual invariant unifying the speed and duration variables. Such an invariant may be related to the integral of speed over time for the entire duration, i.e., the distance made by the stimulus. I suggest that the authors will refer to such a possibility, possibly in light of previous relevant works, if such exist, and possibly by relevant additional analysis of their data in this direction. In any case, a discussion of this possibility is called for.

Reviewer #2: Dear all,

I have read with great interest the work by Toso and collagues. Generally speaking the work is quite sound (save from some issues that I will outline below), however, I have some strong doubts that this paper fits well with Plos Cmputational Biology, a prestigious journal devoted to novel aspects of neuronal computation. Let me be clear, I am not implying that the ms contains no "NEW" material. There is of course a lot of new stuff, but what is missing is that "unexpected-ness" factor. Nor the article is filling a hot gap in the litarature. Of course I may be wrong with my assessment and I ultimately leave it up to the editors but let me line out why I think this is the case.

The work first documents a mutual intereference between vibration intensity and duration estimation in humans and rats. This is new but somewhat doesn't warm the reader much. Indeed the authors themselves acknowledge this in the previous editorial correspondence as well as citing much previous litarature.

The second part of the manuscript then moves onto analisys of activity in somatosensory cortex. The authors report that some neurons can code reliably vibration intensity (sp-coding), some don't. Then they hypothesize two independent read-out/accumulator mechanisms and fine tune them in order to mimic the behaviour of duration judgments and vibration intensity judgments. The result is that one need to postulate rather different mechanisms depending on which is the information that has to be extracted from the neurons. The results are quite sound, however they are not conclusive, in my view. Of course a working model is better than no model at all, however what I am missing is the fact that a computational model (just like any theory) should be able to cast some predictions. And here this part is missing.

Finally they add another psychophysical experiment which demonstrates that the entertwinement between duration and intensity occurs even when subjects know in advance what will be the target judgment and that there is no increase in cross talk between the two dimensions if the subject is forced to hold in memory the two features and is told what to judge only after stimulus presentation. This experiment is presented as a proof of the previous model but in fact it addresses a specific independent point on the nature of this duration-intensity cross talk. In my view this feature is quite independent to the modelling work presented here. Of course I do understand that the computational framework presented here is that of a pool of encoding neurons with two independent read-out/accumulator stage, and soemwhat it is good that experiment 4 revealved the independence from working memory. Yet the results of experiment 4 could also be consistent with two entirely independent (and automatically triggered) neuronal populations. So I don't see how Experiment 4 is a proof of the model validity.

Thus my overall feeling is that the paper is neither resolving an important quarrel in the literature, nor it is offering a compelling explanation for a realtively known phenomenon. It does offer some insights, I am not denying that but it doesn't quite pop.

Ths being said I will line out some specific points on the manuscript

1- Speed and Intensity are two labels chosen to represent the amount of physical vibration and its perceptual counterpart. However this choice is unfortunate. First because the two terms are lexically different. It is as if one spoke about Radiation frequency and Color (or molecular vibration and temperature). Second because outside of this paper, allude to different concepts (speed brings to mind continuous motion, not vibration - intensity brings to mind amount of energy in time). I agree that one does not need to surrender to lay terms, but some effort should be done to offer a smoother taxonomy. One option is to use "average vibration" instead od speed and "vibration intensity" for intensity.

2- The prsentation of the draft leaves something to be desired. It contains inline mention of about 15 figures, yet each figure needs to be downloaded separately. More importantly some crucial aspects are deposited in supplementary material (i.e. the design matrix S1 and the swap of parameters S8) which belong much to the narrative of the paper. I don't know the space constraints of the submission but suggest to revise this. In general it would recommend to trim down the number of supplementary material to offer a better flow.

3- the paper neglects an important phenomenon (central tendency) which predicts non-linear mapping in explicit tasks (such as those of experiment 2). The issue is that with longer durations (and in general with higher noise) there is more tendency to go towards the center of the response line and this predicts a seemingly logarithmic behaviour. (see here https://www.biorxiv.org/content/10.1101/450726v1.full but also the original paper by Jazayeri et al, and follow ups by John Weadern or David Burr). This is particularly relevant here as the authors leverage on this apparent non-linearity to introduce the model (Lines 345-347). Also, I believe that assuming a later stage which bends response mappings, the parameters of the model might turn out to be different. Thus it would be good to include a reference to this point, consider the point that "non linear mapping" does not equate to "non linear encoding", and perhaps discuss how much such later non-linear stages may impact on model parameters.

4- It wouldn't hurt if the authors attempted to cerate a verison of the model which does not exhibit the crosstalk of duration and vibration intensity. I know this is somewhat trivial but I think it would help the reader to see what's peculiar about the very parameters that enable the fit of the psychometric curves. In particular I have the impression that "short time constants for intensity" and "long time constants for duration" are somehow a general property of a good readout mechanism regardless of the cross talk. So overall I think it would be interesting to present results of the fitting procedure aslo in comparison to a model which performs unbiased estimates of duration and intensity

5- In general I think the paper would have gained much more relevance if the authors managed to explain also other phenomena of cross-talk such as those cited or, for instance the tau-kappa effects in vision (space-duration interference). I know this possibly means another paper, but the lack of such breadth of scope is what makes the current submission a bit too narrow in scope and, in my opinion, not up for Plsc Comp Bio

6- Figure 5G is not described in the caption. Also the result is not discussed extensively. Why would "few and noisy neurons" support duration data? And why that only in duration rats? Does this imply that traning is required in order for these neural decoders to emerge?

7- Whenever claims of independence are made, see for instance results of exp4 - Figure 6, Bayes factors should be employed.

8- I am not too fond of the usage of the verb "to accord" So I suggest these changes:

Line 614- Accord -> Fit(s)

Line 641- Accordance -> Agreement

**Have all data underlying the figures and results presented in the manuscript been provided?**

Reviewer #1: Yes

Reviewer #2: Yes

PLOS authors have the option to publish the peer review history of their article (what does this mean?). If published, this will include your full peer review and any attached files.

Reviewer #1: No

Reviewer #2: No
---

## [Editor Report · Decision Letter 1]

4 Jan 2021

Dear Professor Diamond,

We are pleased to inform you that your manuscript 'A sensory integration account for time perception' has been provisionally accepted for publication in PLOS Computational Biology.

Best regards,

Samuel J. Gershman

Deputy Editor

PLOS Computational Biology

---

## [Editor Report · Acceptance letter]

24 Jan 2021

PCOMPBIOL-D-20-01840R1 

A sensory integration account for time perception

Dear Dr Diamond,

I am pleased to inform you that your manuscript has been formally accepted for publication in PLOS Computational Biology. Your manuscript is now with our production department and you will be notified of the publication date in due course.

With kind regards,

Alice Ellingham
